# Deterministic processes structure bacterial genetic communities across an urban landscape

J.M. Hassell [1,2], M.J. Ward[3,4], D. Muloi[2,3,5], J.M. Bettridge[1,2], H. Phan[4,6], T.P. Robinson[7], A. Ogendo[2], T. Imboma[8], J. Kiiru[9], S. Kariuki[9], M. Begon[10], E.K. Kang'ethe[11], M.E.J. Woolhouse [3,5] & E.M. Fèvre[1,2]

Land-use change is predicted to act as a driver of zoonotic disease emergence through human exposure to novel microbial diversity, but evidence for the effects of environmental change on microbial communities in vertebrates is lacking. We sample wild birds at 99 wildlife-livestock-human interfaces across Nairobi, Kenya, and use whole genome sequencing to characterise bacterial genes known to be carried on mobile genetic elements (MGEs) within avian-borne *Escherichia coli* ($n = 241$). By modelling the diversity of bacterial genes encoding virulence and antimicrobial resistance (AMR) against ecological and anthropogenic forms of urban environmental change, we demonstrate that communities of avian-borne bacterial genes are shaped by the assemblage of co-existing avian, livestock and human communities, and the habitat within which they exist. In showing that non-random processes structure bacterial genetic communities in urban wildlife, these findings suggest that it should be possible to forecast the effects of urban land-use change on microbial diversity.

[1] Institute of Infection and Global Health, University of Liverpool, Leahurst Campus, Chester High Road, Neston CH64 7TE, UK. [2] International Livestock Research Institute, 30709 Nairobi, Kenya. [3] Centre for Immunity, Infection and Evolution, University of Edinburgh, EH9 3FL Edinburgh, UK. [4] Nuffield Department of Clinical Medicine, University of Oxford, John Radcliffe Hospital, OX3 9DU Oxford, UK. [5] Usher Institute of Population Health Sciences & Informatics, University of Edinburgh, EH16 4UX Edinburgh, UK. [6] Faculty of Medicine, NIHR BRC Southampton, University of Southampton, SO16 6YD Southampton, UK. [7] Food and Agriculture Organization of the United Nations, 00153 Rome, Italy. [8] National Museums of Kenya, 40658 Nairobi, Kenya. [9] Kenya Medical Research Institute, 54840 Nairobi, Kenya. [10] Institute of Integrative Biology, University of Liverpool, Liverpool L69 7ZB, UK. [11] University of Nairobi, 29053 Nairobi, Kenya. Correspondence and requests for materials should be addressed to J.M.H. (email: hassell.jm@gmail.com) or to E.M.F. (email: Eric.Fevre@liverpool.ac.uk)

Deterministic (i.e., non-random) processes play a central role in shaping how species communities interact with one-another and their environment[1]. As one such process, urbanisation is characterised by extreme habitat fragmentation, which can have profound impacts on the distribution of host populations and epidemiology of infectious disease. In developing cities such as Nairobi, where urban livestock-keeping is commonly practiced as a result of growing demand for animal-sourced food products[2], wildlife frequently co-exist with humans and livestock, forming interfaces across which infectious diseases can pass[3,4]. Changes in the composition and distribution of these host assemblages likely have important implications for microbial epidemiology, determining how pathogens are distributed within their reservoir, and dictating opportunities for spillover into non-reservoir hosts (such as humans)[5–7]. However, there is little empirical evidence that directly links changes in the function of abiotic and biotic systems to the structure of host communities, and dynamics of microbes living within them. Detecting the processes underlying the structure of microbial communities in wildlife and domestic animal populations would bring us a step closer to developing a predictive framework for pathogen emergence at urban wildlife-livestock-human interfaces[8].

Recent advances in sequencing technology, such as whole-genome sequencing (WGS), offer the potential to study the community of genes carried on mobile genetic elements (MGEs) within prokaryote genomes. MGE-borne genes can be horizontally transferred between organisms via recombination mechanisms, and may confer adaptive functional traits such as antimicrobial resistance (AMR) and virulence[9]. The distribution of MGE-borne genes amongst bacteria can therefore provide insight into the community structure of these micro-organisms, an approach that has been successfully used in conjunction with typing tools and time-scaled evolutionary analyses to infer bacterial transmission between hosts[10–12]. The wealth of genetic data generated by WGS could therefore provide an optimal approach to identify key drivers (such as land-use change) that influence the structure of bacterial populations at high risk wildlife-livestock-human interfaces, and assist in untangling the complexity of epidemiological processes, regardless of the taxonomic distance between hosts.

In this study, we apply principals from evolutionary ecology and molecular epidemiology to investigate whether urban ecological processes (e.g., changes in habitat structure and wildlife communities) and anthropogenic processes (e.g., characteristics of human populations, such as density and livestock keeping) occurring across the city of Nairobi, Kenya, are associated with non-random structuring of wildlife-borne bacterial genetic communities. We consider the diversity of MGE-borne genes as a proxy for the diversity of microbial communities within hosts, with the view that the ability of such genes to move relatively freely between bacterial cells through horizontal gene transfer mimics, to an extent, the movement of directly transmitted pathogens between hosts. For commensal bacteria, determinism would be expected in two classes of MGE-borne genes: those encoding AMR and virulence traits, each of which would be expected to respond differently to urban environmental change. Contamination of the external environment with AMR bacteria excreted from humans and livestock treated with antimicrobials (e.g., through sewage effluent or faeces), is considered an important route of wildlife exposure to AMR[13]. As such, if wildlife-borne bacteria are under higher selective pressure to adopt genes encoding AMR in urban areas where greater volumes of antibiotics are consumed, and antibiotic use is more widespread[14,15], the community structure of MGEs encoding AMR would be hypothesised to respond to changes in human activity and the presence of livestock, rather than natural processes

occurring in wildlife communities. In contrast, the diversity of genes encoding virulence traits (for which wildlife-borne bacteria are assumed not to be subjected to such strong anthropogenic selection pressure) would be hypothesised to reflect changes in wildlife host community structure – following the broadly accepted principal that host and microbial community diversity are correlated[5,16], as wildlife host species diversity increases, the pool of virulence-associated MGEs to which they are exposed to should become more diverse.

Adopting the null hypothesis that communities of wildlife-borne bacterial genes are structured by random processes, we test the above expectations by considering variation in the diversity of MGE-borne virulence and AMR genes in commensal *Escherichia coli*, collected from wild birds in household compounds across Nairobi. As likely points of contact (and thus microbial transmission) between vertebrate wildlife, livestock, and humans, household interfaces are chosen as sampling units representative of complex multi-host communities that are widely distributed across a gradient of urban environmental change, and thus suitable for testing our hypotheses. Wild birds are chosen as wildlife hosts in this urban study system, since diverse avian communities distribute widely across urban landscapes[17], demonstrating epidemiological and ecological responses to land-use change[17,18], and interacting closely with livestock and humans[19]. Aside from investigating processes underlying determinism in bacterial genetic diversity, studying the diversity of two sets of genes which may confer adaptive traits to bacteria will enable us to assess whether an association exists between urban land use and the genetic determinants of bacterial selection, with potential implications for human and animal health[9].

## Results

**Bacterial population structure in avian hosts**. Faecal samples ($n = 547$) were collected from 57 avian species in 99 households across Nairobi, that were participating in the UrbanZoo project[20]. Households were selected in such a way that they captured variation in urban land use, wildlife assemblages, human demographics, and livestock-keeping practices across the city (Supplementary Figure 1). A total of 274 *E. coli* isolates, each of which originated from a different individual avian host, were sequenced. Once sequenced, twenty three isolates were removed for being non-*E. coli*, and ten potentially mixed isolates were removed for having a genome size larger than 6 megabases. As such, a total of 241 *E. coli* WGS were considered in further analyses. Genes carried on MGEs, which were known to encode virulence ($n = 63$) or AMR ($n = 47$), were identified in 98% ($n = 236$) and 44% ($n = 107$) of these *E. coli* respectively. *E. coli* population structure across hosts was explored using multi-locus sequence typing (MLST). 128 unique sequence types (STs) were identified, representing a high genetic diversity of *E. coli* in avian samples across the city (Supplementary Figure 2). No sequence type was assigned to 18 isolates that carried at least one novel allele not included in the (MLST) database. The most common STs (ST10, ST155 and ST48; those appearing in > 5% of isolates) were randomly distributed across host functional groups, and not associated with the diversity of MGE-borne AMR and virulence genes in each isolate (Fisher's Exact test: $p = 0.18$; Kruskal–Wallis test AMR genes: $X^2 = 7.17$, $P = 0.62$, $df = 9$; Kruskal–Wallis test virulence genes: $X^2 = 10.4$, $P = 0.11$, $df = 6$).

To test whether microbial genetic communities in avian hosts were deterministically structured in association with the environmental conditions and structure of host communities at household interfaces within which avian hosts resided, the α-diversity of each set of genes (counts, thus representing richness of virulence or AMR genes) was calculated for individual hosts, and

regressed against ecological and anthropogenic characteristics of households using generalised linear mixed effects models (GLMMs). Ecological and anthropogenic factors that were selected as indicators of variation in household environmental conditions, and used as fixed effects in the models, included: α-diversity (richness) of avian species present, biotic habitat diversity, artificial land-use cover (%), wealth indices, livestock-keeping status of each household, livestock density, and human density. Variation in bacterial genetic diversity introduced by differences in the feeding ecology and ranging behaviour of avian hosts was accounted for by including membership of avian hosts to epidemiologically relevant functional groups, and allometrically scaled estimates of each species home range, as fixed effects in each model. Two isolates for which host identity could not be confirmed were excluded from the statistical analyses (bringing the total number of genomes on which analyses were performed to $n = 239$).

**Virulence gene diversity, avian host communities and habitat.** We found that the diversity of virulence genes present in birds varied between host functional groups, and increased with α-diversity of household avian communities (marginal $R^2$: 0.08, Table 1). However, the relationship between virulence gene and avian diversity varied between functional groups, with a significant positive relationship only being present in invertebrate-eating birds (Fig. 1). Habitat diversity and livestock density showed significant inverse relationships with virulence gene diversity (GLMM: $\beta = -0.65$, 95% CI $= -1.17 - -0.13$, $P < 0.05$; GLMM: $\beta = -0.69$, 95% CI $= -1.34 - -0.07$, $P < 0.05$). To further explore determinants of virulence gene diversity in seed-eating birds (which, as synanthropic species, constituted the largest and most well-distributed avian functional group), a separate Poisson-distributed GLMM was built considering only the genetic diversity of sequences derived from this functional group ($n = 152$). This also had the effect of removing variation associated with functional group membership. Once other functional groups had been excluded, habitat diversity had a significant inverse relationship with diversity of virulence genes in seed-eating birds; as habitat diversity decreased, diversity of virulence genes in seed-eating birds increased (GLMM: $\beta = -0.76$, 95% CI $= -1.3 - -0.23$, $P < 0.01$; marginal $R^2$: 0.06).

**AMR gene diversity and assemblages of livestock and humans.** Determinants for the diversity of genes encoding AMR were investigated in a similar way, utilising the same set of avian *E. coli* isolates and household explanatory variables used for virulence genes. The best-fitting model was a zero-inflated hurdle model (with a truncated Poisson error distribution), in which the presence or absence of AMR genes (the zero-inflated component) and increasing diversity of AMR genes (the conditional component) were modelled separately. The conditional model demonstrated that α-diversity of AMR genes was significantly associated with increasing human density, but only in households keeping livestock (GLMM: $\beta = 0.99$, 95% CI $= 0.34 - 1.65$, $P < 0.01$; Table 1). This was supported by the zero-inflated component, which showed a significant negative association between the probability of AMR genes not being detected in avian-borne *E. coli* and increasing human density (GLMM: $\beta = -2.11$, 95% CI $= -3.83 - -0.45$, $P < 0.05$; Table 1). To test whether the interaction between human density and livestock keeping was dependent upon avian host functional-group membership, the same model was fitted independently for isolates derived from seed-eating ($n = 152$) and non-seed-eating birds. This indicated that the relationship between AMR gene diversity, livestock keeping and human density was only present for seed-eating

birds (GLMM: $\beta = 0.91$, 95% CI $= 0.17 - 1.65$, $P < 0.05$; Fig. 2), and that the likelihood of detecting AMR genes increased with the presence of livestock, and increasing human density (Table 1). To explore these relationships further, the fixed covariate livestock keeping was replaced with livestock density (correlation prevented both from being fitted in the same model). The resulting model showed a positive, although non-significant, association between livestock density and diversity of AMR genes in seed-eating birds (GLMM: $\beta = 0.53$, 95% CI $= -0.07 - 1.13$, $P = 0.08$; Table 1).

**Gradients of microbial genetic diversity across Nairobi.** Microbial genetic diversity was framed against city-wide variation in host community structure at household interfaces, by relating the outcomes of our models to the results of an unconstrained principal components analysis (PCA) that was used to decompose variance attributed to avian diversity, livestock density, and human density within households across Nairobi. The first principal component (PC1) accounted for 72.9% of variation, clearly separating households with high avian diversity from households with high human and livestock density. Relating city-wide trends in host community structure to associations between diversity of virulence genes and avian diversity, and diversity of AMR genes and livestock and human density, reveals opposing epidemiological gradients of bacterial genetic diversity across Nairobi (Fig. 3a).

**Discussion**

Understanding the influence of environmental change on the diversity and distribution of microbial communities in wildlife is of fundamental importance to understanding how zoonotic diseases spillover into humans. Here, spatially explicit data on land use, the ecology of host populations, and high resolution microbial sequencing in individual hosts, is linked to explore this question across a developing city. We found that deterministic forces, both ecological (wildlife species assemblages and biotic habitat diversity) and anthropogenic (human and livestock density), operating across the urban landscape of Nairobi are associated with variation in the structure of bacterial genetic communities within avian host communities.

For virulence genes, the species richness of host communities was positively correlated with the diversity of genes present in *E. coli* isolates, with increases in avian diversity being associated with a higher diversity of virulence genes within their *E. coli*. This follows an expected pattern for communities of hosts and their microbial diversity. Assuming each vertebrate host harbours at least some *E. coli* bearing unique virulence genes, increasing vertebrate species diversity will increase the diversity of virulence genes circulating in the population[21] (reviewed by Ostfeld & Keesing[16]). Our results are consistent with the hypothesis that, in this study system, increased vertebrate diversity results in avian-borne *E. coli* acquiring a greater diversity of virulence genes, because of exposure to a larger pool of available genes in the vertebrate host community. The composition and size of this pool of available genes would be hypothesised to vary across a gradient of urban land use, as the structure of avian communities change in response to the changes in habitat structure and biotic resource provision. However, our results also suggest that the relationship between microbial and host community diversity is subject to variation in host functional ecology. For frugivorous birds, which had higher mean diversities of virulence genes, virulence gene diversity was negatively correlated with avian diversity, perhaps because their exposure to *E. coli* harbouring novel virulence genes is driven by dietary exposure rather than transmission between hosts.

**Table 1 Estimated regression parameters, standard errors, z-values and *P* values for optimal generalised linear models used in this study, modelling the diversity of avian-borne *E. coli* virulence and antimicrobial resistance (AMR) genes against household environmental variables**

| Model Terms | Estimate | Std. Error | z value | P value |
|---|---|---|---|---|
| Model 1: Virulence genes, All avian functional groups | | | | |
| Intercept | 1.0542 | 0.4432 | 2.379 | <0.05 |
| Avian Species Richness | 0.0447 | 0.0224 | 2 | <0.05 |
| Fruit/Nectar | 2.321 | 0.831 | 2.793 | <0.01 |
| PlantSeed | 0.5413 | 0.4009 | 1.35 | 0.18 |
| Omnivore | 0.7884 | 0.5745 | 1.373 | 0.17 |
| Livestock Density | −0.6939 | 0.3202 | −2.167 | <0.05 |
| Habitat Diversity | −0.6465 | 0.2598 | −2.488 | <0.05 |
| Avian Species Richness:Fruit/Nectar | −0.1202 | 0.0433 | −2.778 | <0.01 |
| Avian Species Richness:Seedeater | −0.0417 | 0.0236 | −1.768 | 0.08 |
| Avian Species Richness:Omnivore | −0.0478 | 0.0325 | −1.468 | 0.14 |
| Model 2: Virulence genes, Seed-eating birds only | | | | |
| Intercept | 1.8383 | 0.2152 | 8.54 | <0.001 |
| Habitat Diversity | −0.7587 | 0.2698 | −2.812 | <0.01 |
| Livestock Density | −0.6564 | 0.3365 | −1.95 | 0.05 |
| Model 3: AMR genes, All avian functional groups (Zero-inflated hurdle, truncated Poisson) | | | | |
| Conditional model | | | | |
| Intercept | 1.9068 | 0.1562 | 12.204 | <0.001 |
| Livestock kept within household | −0.3171 | 0.1704 | −1.86 | 0.063 |
| Human Density | −0.41413 | 0.2601 | −1.593 | 0.111 |
| Livestock-keeping:Human Density | 0.9948 | 0.3332 | 2.986 | <0.01 |
| Zero-inflation model | | | | |
| Intercept | 1.3175 | 0.4637 | 2.841 | <0.01 |
| Livestock kept within household | −0.8217 | 0.5202 | −1.58 | 0.114 |
| Human Density | −2.1407 | 0.8627 | −2.481 | <0.05 |
| Livestock-keeping:Human Density | 0.3796 | 1.3059 | 0.291 | 0.771 |
| Model 4: AMR genes, Seed-eating birds only (1) (Zero-inflated hurdle, truncated Poisson) | | | | |
| Conditional model | | | | |
| Intercept | 1.8531 | 0.2073 | 8.938 | <0.001 |
| Livestock kept within household | −0.2788 | 0.2227 | −1.252 | 0.211 |
| Human Density | −0.3355 | 0.304 | −1.104 | 0.2698 |
| Livestock-keeping:Human Density | 0.9107 | 0.3768 | 2.417 | <0.05 |
| Zero-inflation model | | | | |
| Intercept | 1.3989 | 0.6002 | 2.323 | <0.05 |
| Livestock kept within household | −1.4189 | 0.6706 | −2.116 | <0.05 |
| Human Density | −2.2329 | 0.9883 | −2.259 | <0.05 |
| Livestock-keeping:Human Density | 1.8476 | 1.4079 | 1.312 | 0.1894 |
| Model 5: AMR genes, Seed-eating birds only (2) (Zero-inflated hurdle, negative Binomial) | | | | |
| Conditional model | | | | |
| Intercept | 1.58251 | 0.0923 | 17.195 | <0.001 |
| Livestock Density | 0.53141 | 0.30462 | 1.745 | 0.081 |
| Zero-inflation model | | | | |
| Intercept | 0.2055 | 0.2295 | 0.895 | 0.371 |
| Livestock Density | −1.1402 | 0.9043 | −1.261 | 0.207 |

For two-stage hurdle models (Models 3–5), a positive contrast in the conditional model represents a higher abundance, whilst a positive contrast in the zero-inflated model indicates a higher chance of absence

The results of this study also indicate that differences in the response of wildlife species to changes in urban land use could play a part in determining how microbial genetic diversity is related to host community diversity. For example, the diversity of virulence genes in *E. coli* derived from seed-eating birds, which show more synanthropic behaviour than other functional groups, was predicted by changes in biotic habitat diversity rather than avian community diversity: in seed-eating birds, increasing virulence gene diversity was linked to decreasing biotic complexity of habitats. Further evidence for the role of host taxa in shaping the response of microbial genetics to variation in urban land use was provided by considering MGEs conferring AMR in *E. coli*. Increasing human and livestock density were associated with higher AMR gene diversity in avian-borne *E. coli*, but this only applied to isolates recovered from seed-eating birds. Importantly, the significant relationship between AMR gene diversity and human density was only found amongst household which kept livestock, providing evidence to suggest that households may act as an interface for the exchange of genes encoding AMR between livestock and wild birds. Livestock and human density could therefore be responsible for influencing the diversity (or pool) of AMR genes present and/or promoting contact with synanthropic wildlife, resulting in spillover of bacteria and/or their genetic elements from livestock to wildlife within household compounds.

Our findings are important for several reasons. First, they point towards the presence of opposing epidemiological gradients for AMR and virulence genes across the urban landscape, in which communities of mobile microbial genes are correlated with changes in the richness and density of vertebrate host communities (which may be confounded by the ecological traits of the host within which that organism resides) (Fig. 3a). Although the

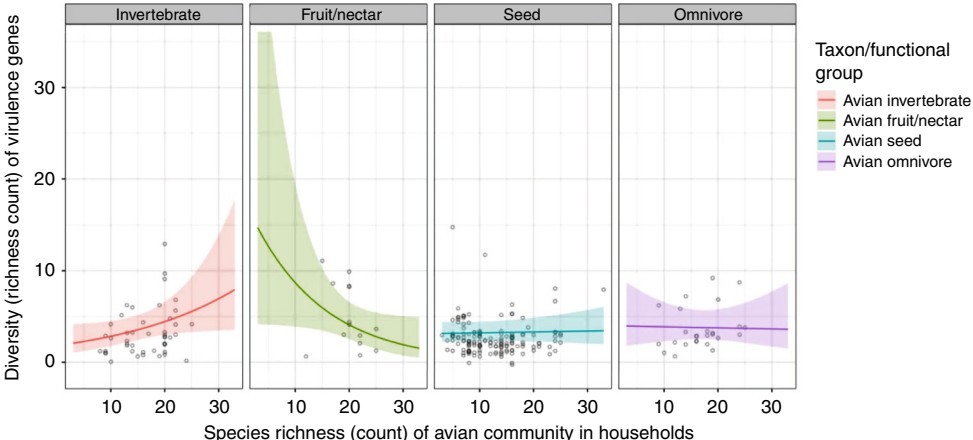

**Fig. 1** Virulence gene diversity and avian species richness. Fit of the Poisson GLMM, modelling how diversity (richness) of virulence genes in avian hosts ($n = 239$) varies as a function of avian host community richness and functional group membership. Coloured lines represent different avian functional groups, and shading on either side of each line represents 95% confidence intervals

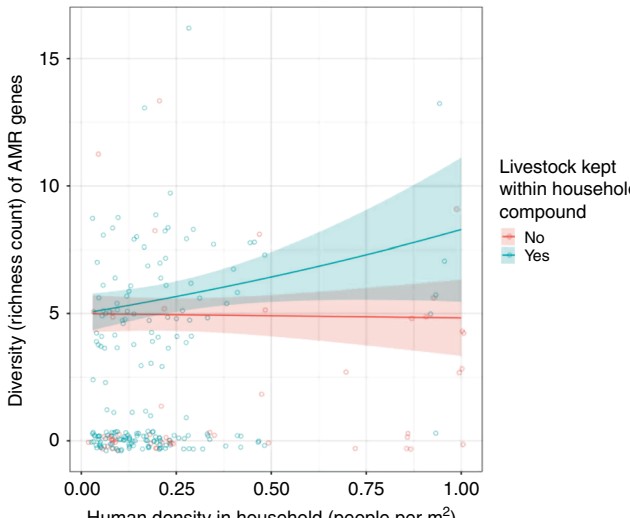

**Fig. 2** Antimicrobial resistance (AMR) gene diversity and human density. Fit of the zero-inflated hurdle model, modelling how diversity of AMR genes in seed-eating birds increased with human density, when livestock were part of the vertebrate host community at household interfaces. Coloured lines represent the presence or absence of livestock in households, and shading on either side of each line represents 95% confidence intervals. A subset ($n = 152$) of *E. coli* isolates were included in this analysis

horizontal exchange mechanisms involved in the transfer of these genes are unlikely to directly mimic the dynamics of microbial transmission, such deterministic patterns might also be displayed by microbial communities subject to the same changes in host community structure. For example, abundance of hosts has been linked to parasite species richness in a number of previous studies[22,23], and increasing diversity of helminth parasitism in Southeast Asian murids has been positively correlated with a gradient of anthropogenic habitat change[24].

Second, our results provide evidence for a mechanism by which anthropogenic processes tied to variation in urban land use result in spillover of MGEs (and potentially microbes) between vertebrate host compartments at wildlife-livestock-human interfaces (Fig. 3b). And third, considering variation in avian community assemblage and the form of human and livestock populations as indicators of differing ecological and anthropogenic processes, our findings suggest that processes associated with urbanisation

can simultaneously exert very different forms of genetic selection (e.g., exposure to diverse pools of virulence or AMR genes) on the same species of bacteria. This could have important implications for public health. For bacterial organisms such as *E. coli*, exposure to larger pools of genetic diversity that promote uptake and fixing of AMR genes can confer adaptive advantages such as drug resistance[25], whilst acquisition of virulence determinents in the accessory genome has been frequently implicated in the emergence of pathogenic lineages of *E. coli*. Divergence associated with horizontal gene transfer between closely related microbial strains can lead to the emergence of novel pathogens[26,27].

In this study, high resolution genetic data collected as part of a structured epidemiological study, was used to study bacterial epidemiology in a multi-host urban system. Whilst the scale of sampling conducted in this study (representing sympatric wildlife, livestock and human communities along a gradient of urban land use) provided the opportunity to explore hypotheses that, until recently could not have been tested, this dataset is not without epidemiological limitations, and the results presented in this study should be interpreted with the following considerations in mind. To better contextualise the transfer of MGE-borne genes, in particular those borne on plasmids, longer read sequencing (e.g., PacBio) would provide an advantage over short-read Illumina data in making epidemiological inferences[28]. However, the focus of this study was on patterns of diversity in terms of gene presence or absence rather than characterising individual genes and the genetic context of their transfer. The sensitivity of commensal *E. coli* in identifying transmission pathways for other pathogens should also be considered with caution. Differences in characteristics (such as shedding rates and effects on host behaviour) between commensal and pathogenic organisms may have epidemiological consequences that reduce their representation of one another. In addition, by only sequencing a single *E. coli* isolate from each host, the within-host genetic diversity of *E. coli* was not considered. Previous molecular studies on *E. coli* (albeit it in different hosts, and using lower resolution sequencing technology), have demonstrated considerable within-host diversity across vertebrate taxa[29–31]. However, the decision to sequence a single isolate from each host was made as a necessary, cost-based trade-off between genetic resolution, depth of sampling *E. coli* genetic diversity within each individual, and the number of unique wildlife individuals from which samples could be included. Under sampling within-host diversity would only be likely to lead to a signal being missed, rather than changes to the positive results that we report in this study.

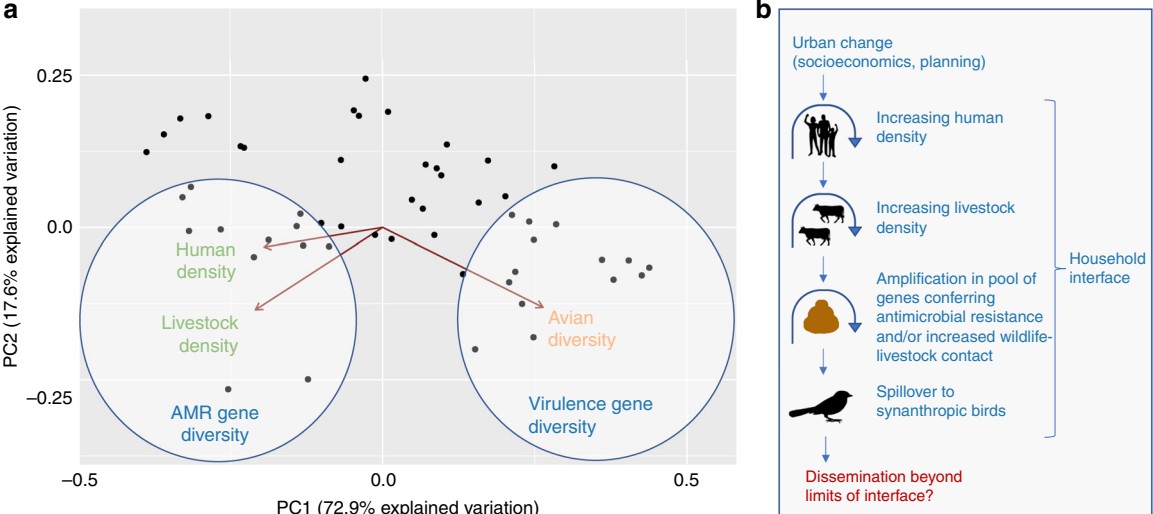

**Fig. 3** Epidemiological gradients in MGE diversity, and pathways to AMR spillover under urban change. **a** Diagrammatic representation of how epidemiological gradients in diversity (richness) of virulence and AMR genes in avian-borne *E. coli* overlay on broad-scale trends in host community characteristics at urban household interfaces. The characteristics of host communities are represented in the form of a principal components analysis (PCA), performed on avian diversity (species richness), human density and livestock density in households. PC1 accounts for most variation, separating households with high avian diversity from households with high human and livestock density. Associations between the diversity of virulence and AMR genes, and avian diversity human density and livestock density are indicated by circles overlaid onto the PCA biplot. **b** Schematic illustrating possible processes leading to spillover of AMR genetic determinants between livestock and synanthropic birds at household interfaces

In demonstrating that it is possible to link epidemiological processes in wildlife to environmental drivers across urban landscapes, this study has taken the first step towards forecasting the effects of urban land-use change on disease emergence within a developing city. Whilst the focus of this study was on wildlife, understanding how urban environmental change structures microbial communities in human and livestock hosts is equally important, and extending analysis of the diversity of genes carried on MGEs to humans and livestock would provide valuable insight into the epidemiological responses of these compartments to variation in land use. By considering genetic diversity in a single species of *Enterobacteriaceae* as a proxy for parasite diversity, this study has necessarily taken a reductionist approach to address important hypotheses that otherwise could not have been answered using this dataset. The limitations in using a model organism such as *E. coli* could be addressed through utilising recent advances in metagenomics, which permit sequencing of bacterial and viral microbiomes, to characterise the structural response of microbial communities to the environmental drivers of urban land-use change. Such methods could be utilised in the future to understand how changes in microbial diversity, and the uptake and fixing of genes by pathogens, translate to emergence and manifestation of clinical disease in wildlife, livestock and humans.

## Methods

**Animal care and use.** The collection of data adhered to the legal requirements of the country in which the research was conducted. Wildlife were trapped under approval of an International Livestock Research Institute (ILRI) Institutional Animal Care and Use Protocol (2015.12).

**Human ethics statement.** Questionnaire data was collected under ILRI Institutional Research Ethics Committee approval (2015-09), and prior informed consent was gained for each individual participating in the project.

**Study design.** The study focused on household livestock keeping as it represents a point of largely unmanaged, intense contact between synanthropic wildlife, livestock and humans. Faecal samples (n = 547) were collected from 57 avian species from 99 households across Nairobi, that were participating in the UrbanZoo project[20]. The UrbanZoo project, based in Nairobi, Kenya from 2012–2017, aimed

to utilise a landscape genetics approach to understanding the movement and sharing of pathogens in a major developing city. A key component of this project, within which this study was nested, was the '99 household project', which focused on informal livestock keeping practices in urban households as a route of zoonotic disease emergence in humans. As such, households were selected with the aim of maximising the spatial distribution and diversity of livestock keeping practices across Nairobi, and were chosen to capture three main criteria: socio-economic diversity, population distribution and livestock keeping practices. Geospatial mapping data, generated as part of a technical report produced by Institut Français de Recherche en Afrique (IFRA), was used to identify 17 classes of residential neighbourhood in Nairobi based on physical landscape attributes, which were subsequently verified by 817 household questionnaires[32]. Each of the 17 classes of neighbourhood were then ranked by average income and reduced into seven wealth groups. Administrative sublocations were mapped onto each wealth group, identifying a total of 70 possible sublocations, for which dominant wealth groups were calculated by extracting the proportion of population belonging to each neighbourhood class within the sub-location boundaries (Supplementary Table 1). A total of 33 sublocations were selected to be included in the study, with the number of sublocations belonging to each wealth group chosen proportionately to the population density and the variety of neighbourhood classes in each of the seven wealth groups. Final selection of individual sublocations was aimed at maximising areas with high livestock densities, whilst ensuring coverage of other neighbourhood classes and geographical spread.

For each sublocation, three geographical points were selected at random within the dominant housing type. The order in which sublocations were visited was randomised. Local officials assisted in the recruitment of a household closest to each geographical point, to obtain two livestock keeping and one non-livestock keeping household per sublocation (a total of 99 households, 66 of which kept livestock). Households had to meet strict inclusion criteria of keeping either large ruminants (cattle), large monogastrics (pigs), small ruminants (goats/sheep), small monogastrics (poultry/rabbits), or no livestock species. To ensure an equal sample of both cattle and pig-keeping households, the combination of livestock keeping households represented in each sublocation was randomised, and had to consist of either large ruminant and small monogastric, or large monogastric and small ruminant species. For sublocations in which households keeping large ruminant or large monogastric species were absent, a replacement household keeping either small monogastic or small ruminant species was recruited. Sampling of households took place between September 2015 and September 2016.

**Wildlife trapping and ecological surveys.** A dedicated field team was responsible for collecting data on humans, livestock and wildlife in each household, consisting of veterinarians, animal health technicians and clinicians. Mist nets were set at dawn to trap birds, with nets being positioned outside the house and around livestock keeping facilities. Once caught, all birds were live-sampled in the field under manual restraint, before being released unharmed. Morphometric data were collected for identification purposes, and a suite of biological samples (including

faeces if available, or a cloacal swab) were collected from each animal. Due to large variation in the size of household compounds, trapping effort (i.e., number of mist nets placed per trapping session) was maintained such that it was proportional to the size of the household compound. Ecological surveys were used, alongside trapping data, to estimate the diversity of avian species present within households. Avian species counts (presence/absence) were conducted by a trained ornithologist from the National Museums of Kenya, in which species were identified based on audio-visual identification over a 20-minute period spent walking transects of each household compound. Surveys were conducted between 6:30am and 9:30am, over the course of two months in the dry season, ensuring that bird activity and weather conditions were constant. The species richness of avian communities (α-diversity: the total number of avian species recorded in a household) was calculated for each household. Avian species were also grouped into five functional groups, deemed relevant for the epidemiology of a directly transmitted gastrointestinal parasite such as *E. coli*; plant/seed-eating, omnivorous, fruit/nectar-eating, invertebrate-eating, vertebrate/fish-eating/scavenger. Allocation of avian species to functional groups was based upon the EltonTraits database[33]. Home range estimates for all avian species were calculated by allometric scaling of body weight[34]. Scaling factors published for functionally different birds by Ottoviani et al.[35] were used, and species mean body weights were either collected during sampling, or sourced from published datasets when unavailable[36].

**Household questionnaires.** A nominated member of each household completed a questionnaire, detailing *i)* livestock ownership, management, sourcing, sales and antimicrobial use, and *ii)* household composition and socio-economic data. Abundance (counts) of livestock species and humans were derived from this data for each household. Dividing livestock and human abundance by household area (meters[2], as measured using ArcGIS) generated an estimate of density of livestock and humans. Household composition and socio-economic data were used to generate wealth and ruralness indices for each household sampled[20]. These indices were calculated based on methods used to create the Demographic and Health Surveys (DHS) wealth index, which is derived from a Principal Component Analysis (PCA) of easily measurable households assets (such as access to water, construction materials and ownership of livestock)[37]. A modification was made to the original set of household assets included in the DHS index to better capture household variation in Nairobi. All field data was recorded using Open Data Kit (ODK) Collect software (Hartung et al., 2010), on electronic tablets, and uploaded to databases held on servers at the International Livestock Research Institute (ILRI).

**Land-use classification.** Nairobi is characterised by a large variety of land use. Land use comprises the biotic and abiotic niches within which hosts exist, and was classified for each household. The boundary of each household compound was drawn in ArcMap, and a 30 m buffer created around the perimeter of each compound to represent the landscape surrounding it. A buffer of 30 m was chosen to reflect home range of common urban rodent species (*Mus* and *rattus* spp., estimates of which vary from 1 m to 30 m)[38,39]. Visual classification of land-use types within the compound and buffer area were conducted at 1:500 scale on a 1 m resolution ESRI World Imagery satellite-image available in ArcGIS 10.5 (ESRI). Characterisation of ecological characteristics along a perimeter around the household compound was considered as important, because the ecological setting within which the household exists extends beyond the boundaries of the compound. The extent to which this influential area of habitat outside the compound extends is unknown, and as such it was standardised across study sites. Within the boundary, the areas of nine different land-use types were visually identified and sketched as polygons; water-body, wetland, crops, mature trees, shrubs, grassland, bare ground, artificial ground and rubbish (descriptions for each of these are summarised in Supplementary Table 2). The total area of classified land-use types at each site were calculated and expressed as proportions. Ecological land-use types (all except bare ground, artificial and rubbish) were used to calculate Simpson's diversity index, which considers both habitat richness, and an evenness of abundance among the land-use types present at each site. This index was created to represent the diversity of living (biotic) habitat niches available to wildlife within households, and ranged from 1 (maximum heterogeneity) to 0 (only a single category of biotic land use present). All classification was undertaken by J.M.H. who was familiar with the landscape at each site, and subsequently ground-truthed by revisiting sites.

**Microbiological testing.** All swabs and fresh faecal samples were placed in Amies transport media and transported on ice to one of two laboratories (Kenya Medical Research Institute (KEMRI) or University of Nairobi (UoN)). Samples were enriched in buffered peptone water for 24 hours, and plated onto eosin methylene blue agar (EMBA). Plates were incubated for 24 hours at 37 °C, after which five colonies were selected from each EMBA plate. After a further sub-culture on EMBA to purify the isolates, the pure isolates were sub-cultured on Müller-Hinton (MH) agar and archived at −80 °C in cryovials containing Soy broth supplemented with 15% glycerol.

**Next-generation sequencing.** A single colony was picked from each original sample (referred to as an isolate) and biochemical tests (triple sugar iron agar, Simmon's citrate agar, and motility-indole-lysine media) were run for identification as *E. coli*. DNA was extracted from bacterial isolates using commercial kits (Purelink® Genomic DNA Mini Kit, Invitrogen, Life Technologies, Carlsbad, California) and transported under licence to The Wellcome Trust Centre for Human Genetics Oxford (MMM) Group to: (i) perform standard quality control checks using fastQC (https://www.bioinformatics.babraham.ac.uk/projects/fastqc/) with default settings; (ii) trim reads to remove remnant adaptor sequences using bbduk[40] (parameters: minoverlap = 12, *k* = 19, mink = 12, hdist = 1, ktrim = r) and (iii) perform a Kraken[41] speciation analysis against with an in-house database of bacterial reads downloaded from the NCBI sequence read archive (www.ncbi.nlm.nih.gov/sra/), with an automated step for removal of contaminant (non-bacterial) reads. De novo assembly was performed using SPAdes v3.6[42] (parameters:–careful, -t 1,–phred-offset 33). The assemblies were run through the batch upload mode of the Centre for Genetic Epidemiology web interface hosted by the Technical University of Denmark (https://cge.cbs.dtu.dk/services/cge/) which performs speciation analysis[43], multilocus sequence typing (MLST)[44], detection of resistance genes[45] and detection of virulence genes[46]. The threshold of AMR gene detection was set to 90% identity and 60% coverage, as this is shown to be the optimal threshold for this method. A 60% coverage threshold was used to ensure that AMR genes spread over two contigs, and/or located on the edge of the contig, were not missed[45]. Virulence genes were identified using VirulenceFinder with 90% minimum match and 60% minimum length. Samples deemed as non-*E. coli* on the basis of the speciation analysis with kmerFinder[47] in the Centre for Genetic Epidemiology pipeline were excluded from further analysis. Potentially mixed *E. coli* samples were identified as those with an unusually large assembly size (greater than 6 megabases (Mb)) and were removed from the dataset. Supplementary Table 3 details the QC and assembly metrics of the 241 *E. coli* isolates included in the study.

**Statistical analyses.** All statistical analyses were conducted using R v3.3.2[48]. The response variables diversity of virulence and AMR genes, were regressed against explanatory variables in generalised linear mixed effects models (GLMMs). Isolates for which AMR or virulence genes were not detected were included in these analyses. To address the fact that genes co-mobilised on the same MGE might not represent independent acquisition events without having access to long-read sequencing (which would enable identification of the location of genes on plasmids), we combined all pairs of genes with 100% co-occurrence (e.g., bfpA and perA). To account for the dependency structure of the data, the household and sublocation in which samples were collected were included as nested random effects. To account for the relationship between bacterial population structure and MGE diversity, we also included a measure of bacterial population structure as a random effect in each model. Due to high MLST diversity in the dataset (128 unique STs, and 18 novel STs), sequence type could not be included as a random effect, and as such, each isolate was assigned to a less stringent cluster using the BURST algorithm, on the basis of 3 rather than 7 genetic loci. This composite measure of genetic structure was included as a random effect in each model. Models of virulence gene diversity were fitted with a Poisson distribution in the R package lme4[49]. Preliminary data exploration indicated substantial zero-inflation in the response variable α-diversity of AMR genes (i.e., many samples where no AMR genes were detected), and as such a zero-inflated Poisson model (ZIP) was initially fitted to the data (56% of data comprising the response variable were zeros). However, residuals from the optimal ZIP model obtained through step-wise selection showed considerable overdispersion (dispersion statistic: 3, a value of 1 is considered to represent adequate statistical dispersion). Dispersion parameters were stabilised by fitting zero-inflated mixture and hurdle models available in the R package glmmTMB[50] to the data. These classes of model are frequently used to model zero-inflated count data in ecological datasets. The fit of these models were compared using Akaike's information criteria (AIC).

Optimal models were constructed using stepwise, backwards elimination from the full model based upon (AIC). Significance of model terms were tested by the maximum likelihood test, and the fit of each model was reported as marginal regression coefficients of multiple determination (marginal $R^2$) where possible. Model assumptions were verified by plotting residuals versus fitted values, and by assessing models for overdispersion. Non-linear relationships were checked by fitting a generalized additive model (GAM) between the response and explanatory variables, featuring a nonlinear smoother, in R package mgcv[51]. The residuals were also assessed for spatial dependency by plotting them against geographic coordinates, and examining the results of a semivariogram.

An unconstrained principal component analysis (PCA), was performed on avian diversity, livestock density, and human density within households across Nairobi, in the R package vegan[52].

**Reporting summary.** Further information on research design is available in the Nature Research Reporting Summary linked to this article.

## Data availability

Data (AMR and virulence gene datasets, and accompanying metadata) are available via an open access repository held by the University of Liverpool (http://dx.doi.org/10.17638/datacat.liverpool.ac.uk/738). All sequencing reads are available on the European Nucleotide Archive, under Project ID: PRJEB32607.

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

## Acknowledgements

The views and opinions expressed in this article are those of the authors and do not necessarily reflect the official policy or position of any agency. We would like to thank the help and support from the Department of Veterinary Services within the Kenya Ministry

of Livestock Development, the National Museums of Kenya and the Kenya Wildlife Service. We thank the Oxford Genomics Centre at the Wellcome Centre for Human Genetics (funded by Wellcome Trust grant reference 203141/Z/16/Z) for the generation and initial processing of the sequencing data. We thank the Oxford NIHR Biomedical Research Centre for the use of computing resources, and Prof Daniel Wilson of the Modernising Medical Microbiology group for helpful discussions about E. coli genetics. We also thank the Wellcome Centre for Infection Immunity and Evolution in Edinburgh. We are grateful to all members of the UrbanZoo field team, who helped collect avian samples and household questionnaire data, and all members of the UrbanZoo laboratory teams, who conducted microbiological culture and testing. Without their help, this study would not have been possible. Finally, we would like to thank the numerous people in households across Nairobi who were interviewed for this project. Funding for this study was provided by the UK Medical Research Council, through the Environmental and Social Ecology of Human Infectious Diseases Initiative (ESEI), a cross research council initiative supported by the Medical Research Council, Biotechnology and Biological Science Research Council, the Economic and Social Research Council and the Natural Environment Research Council (grant reference: G1100783/1), and in part, by the CGIAR Research Program on Agriculture for Nutrition and Health (A4NH), led by the International Food Policy Research Institute; we acknowledge the CGIAR Fund Donors (https://www.cgiar.org/funders/). M.J.W. was supported by a Sir Henry Wellcome Postdoctoral Fellowship from the Wellcome Trust (Grant Reference: WT103953MA).

## Author contributions

E.M.F., T.P.R., S.K., E.K.K., M.E.J.W., M.J.W. and J.M.H., conceived the study. J.M.H. collected field data, performed all data analysis, and drafted the manuscript. J.M.B. was involved in the design of data collection protocols, and collected field data. A.O. and T.I. collected field data. J.K. developed laboratory protocols, and conducted microbiological testing. M.J.W., D.M. and H.P. designed and performed bioinformatic analysis. M.B. contributed intellectually to the study design and analysis. All authors provided comments on the manuscript and gave final approval for publication.

## Additional information

**Competing interests:** The authors have no competing interests.

