## [peer review file · Nature Communications]

Reviewer #1 (Remarks to the Author):

This interesting and relatively novel study attempts to identify how urbanization, along with livestock raising in urban environments, impacts bacterial communities in wild birds living at the wildlife-livestock-human interface. The study evaluates mobile genetic element (horizontally transferred genes in *E. coli*) diversity as a proxy for within-host microbial community diversity, and evaluates the diversity of antimicrobial resistance (AMR) and virulence genes in *E. coli* in 547 (57 species) wild birds captured in 99 urban households located within 33 sublocations where households harbored domestic animals. These urban households were classified by wealth groups and other features, such as tree cover and plot size, socioeconomic status, and livestock-keeping characteristics. The principal results of this study show that 1) *E. coli* virulence gene diversity in wild birds increased with avian diversity (species richness) measured at the 'household' level, except for in seed-eating birds 2) *E. coli* virulence gene diversity increased as habitat diversity decreased, 3) Livestock keeping and human density were associated with increased AMR diversity, particularly in seed-eating birds.

[1]

A major critique that I have of this study is the scale mismatch at which avian community diversity is evaluated, regardless of the significant patterns and association seen. First, avian community structure and diversity (species richness) in this study is measured and defined at the household level, when the true scale at which wild bird communities use the habitats that they live in likely differs from the household scale as measured in this study (even though efforts were made to adjust sampling relative to household size), and the diversity and/or species richness is likely defined by the spatial distribution of available resources, behavior, home range, and foraging behavior (habitat use) of the birds, and may be better defined by evaluating something like vegetation characteristics, species accumulation curves across a larger spatial scale or replicates of collection sites) and resource distribution of the different avian guild species, and if some of these species are habitat generalists or specialists. Wild birds perceive their habitat, and can move and forage, across many different household types and conditions including those with and without livestock, and very few birds probably have a home range within a defined household area, except for the largest household landholding areas in this study. In other words, the wild bird species are likely very incomplete when measured at a household level, and at different points in time at a particular household, there may be a wide range of avian species diversity, that may be particularly high during times 'group foraging' birds are passing through a household yard. A small rodent model, using a mouse or rat with a more household-level home range size would probably be a better 'fit' to explore the questions of impacts of urbanization on pathogen transmission or bacterial community composition at the wildlife-livestock-human interface.

In the abstract, authors state that 'we demonstrate that communities of bacterial genes in avian hosts are shaped by the structure of the co-existing host communities.' I might have missed this, but I do not think that bacterial community structure/characteristics data from co-existing hosts (e.g. domestic animals and humans) was presented in this study,

[2]

Figures comments:

Figure 1- It would be useful to also color code the individual data points for avian guild

Figure 2- It would be useful to color code/or shape code the individual data point regarding the observation of livestock kept in the household compound or not.

Figure 3 is a nice analytic summary of the study findings. The broad-scale findings as a whole are quite interesting.

A map of the study area and households sampled with some coding of the different household and habitat classification/characteristics would allow the reader to look at spatial relationships between the sample sites, and would be useful to include in at least as a supplementary figure.

Minor comments:

Line 45-46- can you specify what you mean by 'epidemiological processes'.

Line 67- change 'urban environmental change' to 'urbanization' and 'ecological fragmentation' to 'habitat fragmentation'

Line 68- change 'distributional ecology' to 'distribution'

Line 72, delete 'epidemiological' and change 'can be passed' to 'can pass'

Line 78 change 'determinism in host communities' to 'the structure of microbial communities in wildlife and domestic animal populations'

Lines 94-95, can you specify what you mean by "urban ecological and anthropogenic processes"

Line 107- change 'Utilising' to 'Using'

Line 118- this statement is unclear, can you please clarify? '

The community structure of MGEs encoding AMR would be hypothesized to respond to anthropogenic changes, rather than ecological variability in host communities'

Line 292, change 'Utilised' to 'used'

Lines 331-334 can be deleted.

Check the reference formatting, as not all of the references have formatted correctly (e.g. Ref #30).

Reviewer #3 (Remarks to the Author):

This is an extremely well written paper with a very rigorous study design that is set out exceptionally well. I really enjoyed reading this paper and found the conclusions especially interesting and novel. this work will be of great interest to a broad audience.

My comments are limited as I thought the authors did an excellent job.

The authors might consider evaluating the potential interaction between AMR and virulence - a growing area of interest. Both have important implications to bacterial fitness and survival, elements that may well be interdependent with potential impacts on microbial population structure.

I also felt that there was need to report and consider the number of birds sampled by species in a particular study compound. While diversity is important, the number of avian hosts sampled for a given species at any given place and time might be expected to influence the probability of detecting MGE-borne genes in *E. coli* isolated from a particular avian species. This potential bias should I think be considered within the structure of the GLMM. Perhaps this issue is dealt with in some way that I did not appreciate. More information here might help.

I think the authors might usefully note that there is some evidence to suggest that AMR genes persist even in the absence of selective pressure – this has important implications to conclusions made. In particular, it might be useful to provide some indication of how far these bird species move? Are any of them migratory? How might this influence your conclusions and limitations of the study?

Minor points

1. Line 231- This sentence is difficult to understand and should be revised.
2. Lin 126 - Do the genes exert selective pressure?
3. Further to #2, while a gene might be detected, it may not be phenotypically expressed – this is something that should be noted.

This is a really splendid piece of work.

Kathleen Alexander

Reviewer #4 (Remarks to the Author):

General statement

Overall this is an interesting paper exploring the impacts of changing land-use as a driver for the emergence of zoonotic disease by modelling the acquisition of virulence and antimicrobial resistance genes in *E. coli* isolated from birds. This is a unique dataset that I believe will be of interest to many. However, I cannot recommend that it is accepted in its current form. Further analyses, particularly of the *E. coli* genomes, and some additional clarification of details would strengthen the stated key findings of this manuscript.

Major comments

1) A key part of the analysis is that the population structure within the *E. coli* isolates was not explored at all. This is important as *E. coli* is a diverse species with different lineages that vary in accessory genome content – a key element of this study. At present, it appears that the *E. coli* isolates are being treated as one largely homogenous group.

- While the MLST was determined for the isolates, this was not reported anywhere in the manuscript. While MLST provides only a crude overview of the population structure, these data should be reported.
- Given the WGS of the *E. coli* isolates are available, the population structure should be inferred using phylogenetic methods commonly used in bacterial genomics.
- It would be of interest to know if the bacterial population structure correlated with host species at all suggesting possible adaptation to different wild birds. The bacterial population structure may impact the statistical modelling and may need to be addressed. If it does not impact the modelling this should also be addressed.

2) Co-occurrence of mobile genetic elements. The co-occurrence of both virulence genes and AMR genes was not explored in this study. This is important as the movement of genes together on mobile genetic elements means not all genes detected may be treated as independent acquisition events which may impact the modelling.

- For example, several of the virulence genes detected are all located on the locus of enterocyte effacement (LEE) pathogenicity island (*tir*, *eae*, *espA*, *espB*, *espF* etc). Was there any effort to account for genes that would likely move together on the same pathogenicity island or plasmid (in the case of *bfpA* and *perA*)?
- The co-occurrence of AMR genes should also be considered. E.g. *strA/strB* mediate resistance to streptomycin and are usually co-mobilised together. Does this impact the statistical modelling? I suspect that it would.
- It may be of interest to explore the presence of different plasmid replicons with common AMR genes. The plasmid replicons were screened for already but the data not reported anywhere.

3) More details of the genomic analyses are required for the virulence and AMR screens which utilised spades-generated assemblies. What was the minimum coverage and minimum identify used? It is important these thresholds are defined. For example, genes in the VF database that are encoded on the LEE were not detected in all isolates. E.g. while *tir* and *espA* were detected in 12 isolates, *eae* was not detected in 2/12 isolates and *espB* was not detected in any of the isolates. This suggests that some of the genes may be missed in the screen possibly due to thresholds applied or perhaps that several of the genes are known to have sequence diversity (there are several papers exploring the diversity of the LEE) and the different variants of these genes may not be in the CGE database. Please comment on these issues.

4) While not the focus of this study, it would be of interest to summarise the AMR patterns detected in these *E. coli* isolates for example, by drug class over the sampling period from 2012-2017. Was there an increase in AMR in general or in any specific class over this sampling period? Are there any data on the use of antimicrobials in Nairobi, Kenya? This would be of broad interest as there is little AMR data from commensal *E. coli* isolates.

5) Fig 1. It was difficult to see the 95% confidence intervals – perhaps split out into four panels to make easier to interpret. The diversity of virulence genes on the y axis – is this based on counts of total number of virulence genes? The same applies for the species richness of avian community in households on the x axis – is this counts of different wild birds detected at a site? Please give *n* for the number of *E. coli* genomes in the analysis and make the x and y axis labels more informative.

6) Fig 2. Please give *n* for the number of *E. coli* genomes in the analysis. Please give further information to interpret the x axis for human density. It is unclear what e.g. 0.5 human density meant.

7) It was unclear what was meant by lines 285-290 and why there was a shift to viruses when the focus of the paper is *E. coli*. The emergence of different pathogenic lineages of *E. coli* have been shown many times through the acquisition of virulence determinants in the accessory genome (e.g. on pathogenicity islands or on plasmids).

8) Please provide further details on the QC and filtering of genomes in Methods (Next-generation sequencing paragraph). Please expand upon standard quality controls checks of the short read data or provide a reference. E.g. did this include using the phred score?

9) It was unclear as what was the final number of *E. coli* genomes included in this study and methods for selecting the final isolates lacked detail. While 242 is given as the number of isolates with either AMR or VF genes detected, it was not clear if this was all the genomes included or a subset of those included.

- Please provide further details and numbers of E. coli genomes in the Next Gen Sequencing section that resulted in the final number. As it currently reads I understand that there were 547 strains subject to WGS.
- Please provide further detail as to how human and virus read removal was undertaken (line 459).
- Please provide details as to how isolates were deemed non-E. coli (lines 461-462). What was the number excluded and on what basis – e.g on mapping results to AE014075.1?
- What was the number of potentially mixed E. coli isolates based upon genome assemblies? It would be useful to provide a supplementary table of the QC and assembly metrics of the final E. coli isolates included in this study.
- Were there isolates that passed all the filtering steps but had no AMR or VF genes detected and were not included in the study? If so, what was the number?
- In the AMR and VF tables provided – there were 241 not 242 isolates. Is the number 241?

11) The short read data and genome assembly data for the E. coli should be made publicly available on a BioProject. This would facilitate others to be able to use and reproduce these data in the future.

12) While the AMR and VF data are available at an open access repository held by the University of Liverpool, the specific data available should be stated in the manuscript as it was not initially clear that the AMR and VF data was available.

Minor comments

- 1) Please add the n value for the number of E. coli isolates in this study to the abstract.
- 2) Please check italicisation of E. coli (line 120) and that E. coli is spelt out in full in the introduction at first use (line 97 and not line 107).
- 3) Please use acronyms after first use (e.g. line 135 whole genome sequences).

Reviewer #1 (Remarks to the Author):

This interesting and relatively novel study attempts to identify how urbanization, along with livestock raising in urban environments, impacts bacterial communities in wild birds living at the wildlife-livestock-human interface. The study evaluates mobile genetic element (horizontally transferred genes in *E. coli*) diversity as a proxy for within-host microbial community diversity, and evaluates the diversity of antimicrobial resistance (AMR) and virulence genes in *E. coli* in 547 (57 species) wild birds captured in 99 urban households located within 33 sublocations where households harbored domestic animals. These urban households were classified by wealth groups and other features, such as tree cover and plot size, socioeconomic status, and livestock-keeping characteristics. The principal results of this study show that 1) *E. coli* virulence gene diversity in wild birds increased with avian diversity (species richness) measured at the 'household' level, except for in seed-eating birds 2) *E. coli* virulence gene diversity increased as habitat diversity decreased, 3) Livestock keeping and human density were associated with increased AMR diversity, particularly in seed-eating birds.

A major critique that I have of this study is the scale mismatch at which avian community diversity is evaluated, regardless of the significant patterns and association seen. First, avian community structure and diversity (species richness) in this study is measured and defined at the household level, when the true scale at which wild bird communities use the habitats that they live in likely differs from the household scale as measured in this study (even though efforts were made to adjust sampling relative to household size), and the diversity and/or species richness is likely defined by the spatial distribution of available resources, behavior, home range, and foraging behavior (habitat use) of the birds, and may be better defined by evaluating something like vegetation characteristics, species accumulation curves across a larger spatial scale or replicates of collection sites) and resource distribution of the different avian guild species, and if some of these species are habitat generalists or specialists. Wild birds perceive their habitat, and can move and forage, across many different household types and conditions including those with and without livestock, and very few birds probably have a home range within a defined household area, except for the largest household landholding areas in this study. In other words, the wild bird species are likely very incomplete when measured at a household level, and at different points in time at a particular household, there may be a wide range of avian species diversity, that may be particularly high during times 'group foraging' birds are passing through a household yard. A small rodent model, using a mouse or rat with a more household-level home range size would probably be a better 'fit' to explore the questions of impacts of urbanization on pathogen transmission or bacterial community composition at the wildlife-livestock-human interface.

JH response: We agree with the reviewer that birds' ambits would be expected to extend beyond what we call the 'household scale'. Cities such as Nairobi are characterised by localised patches of similarly structured habitats. These patches are reasonably well captured by the division of the city into sublocations. We expect avian richness (which, as has been shown in other studies, is influenced by structural vegetation diversity and resource provisioning in urban environments <https://link.springer.com/article/10.1007/s11355-011-0153-4>) to scale to this level of urban environmental organisation.

Our study was explicitly designed to capture variation in sympatric wildlife, livestock and human communities at the scale of sublocation. Sublocations from across the city were

selected through stratified random sampling of wealth groups (a major driver of habitat). Triplets of households purposefully selected within sublocations therefore represent multiple samplings of patches of similar habitat (see the maps in Supplementary Figure 1 reflecting spatial distribution and habitat characteristics of households), at a scale we believe to be appropriate for most wild bird species. The variance at this scale is accounted for in our analysis by including sublocation as a random effect in the statistical models.

We also agree with the reviewer that small rodents could also act as a good model with which to study bacterial community composition at the household wildlife-livestock-human interface. *E. coli* was isolated from rodents trapped in households as part of the UrbanZoo project, but not in sufficient numbers for statistical analyses to be conducted. In addition, the extensive effort required to accurately estimate rodent diversity across a wide variety of household ‘habitats’ was outside the scope of our study.

In the abstract, authors state that ‘we demonstrate that communities of bacterial genes in avian hosts are shaped by the structure of the co-existing host communities.’ I might have missed this, but I do not think that bacterial community structure/characteristics data from co-existing hosts (e.g. domestic animals and humans) was presented in this study,

JH response: This sentence relates to the structure of the animal and human host communities at the scale at which avian hosts were sampled. In this study, this host community structure is represented as avian species richness, and livestock and human density. To clarify this we have changed the wording of this sentence to “we demonstrate that communities of bacterial genes in avian hosts are shaped by the assemblage of co-existing avian, livestock and human communities”

Figures comments:

Figure 1- It would be useful to also color code the individual data points for avian guild

JH response: Thank you for this suggestion. We have reformatted the plot so that each avian functional group is plotted on a separate graph.

Figure 2- It would be useful to color code/or shape code the individual data point regarding the observation of livestock kept in the household compound or not.

JH response: Thank you for this suggestion. We have reformatted the plot in Figure 2 so that data points are coloured by the presence of livestock within households.

Figure 3 is a nice analytic summary of the study findings. The broad-scale findings as a whole are quite interesting.

A map of the study area and households sampled with some coding of the different household and habitat classification/characteristics would allow the reader to look at spatial relationships between the sample sites, and would be useful to include in at least as a supplementary figure.

JH response: We appreciate that a map of the study area, featuring household habitat characteristics would be informative to the reader. As such, we have created a Supplementary Figure (Supplementary Figure 1), which depicts the proportion of different land use types across the households that participated in the study.

Minor comments:

Line 45-46- can you specify what you mean by 'epidemiological processes'.

JH response: We have elaborated on this, so that the sentence now reads – "...we demonstrate that it is possible to link basic epidemiological processes occurring in natural host communities to ecological and anthropogenic drivers across an urban landscape."

Line 67- change 'urban environmental change' to 'urbanization' and 'ecological fragmentation' to 'habitat fragmentation'

JH response: These changes have been made.

Line 68- change 'distributional ecology' to 'distribution'

JH response: This change has been made.

Line 72, delete 'epidemiological' and change 'can be passed' to 'can pass'

JH response: These changes have been made.

Line 78 change 'determinism in host communities' to 'the structure of microbial communities in wildlife and domestic animal populations'

JH response: This change has been made.

Lines 94-95, can you specify what you mean by "urban ecological and anthropogenic processes"

JH response: We have made this clearer, and this sentence now reads:

"...to investigate whether urban ecological processes (e.g. changes in habitat structure and wildlife communities) and anthropogenic processes (e.g. characteristics of human populations, such as density and livestock keeping) occurring across the city of Nairobi, Kenya, are associated with non-random structuring of wildlife-borne bacterial genetic communities"

Line 107- change 'Utilising' to 'Using'

JH response: This change has been made.

Line 118- this statement is unclear, can you please clarify? 'The community structure of MGEs encoding AMR would be hypothesized to respond to anthropogenic changes, rather than ecological variability in host communities'

JH response: We have made this clearer, and this sentence now reads:

"...the community structure of MGEs encoding AMR would be hypothesised to respond to changes in human activity and the presence of livestock, rather than natural processes occurring in wildlife communities"

Line 292, change 'Utilised' to 'used'

JH response: This change has been made.

Lines 331-334 can be deleted.

JH response: This change has been made.

Check the reference formatting, as not all of the references have formatted correctly (e.g. Ref #30).

JH response: Thank you for highlighting this – the reference list has been checked and corrected where necessary.

Reviewer #3 (Remarks to the Author):

This is an extremely well written paper with a very rigorous study design that is set out exceptionally well. I really enjoyed reading this paper and found the conclusions especially interesting and novel. this work will be of great interest to a broad audience.

My comments are limited as I thought the authors did an excellent job.

The authors might consider evaluating the potential interaction between AMR and virulence - a growing area of interest. Both have important implications to bacterial fitness and survival, elements that may well be interdependent with potential impacts on microbial population structure.

I also felt that there was need to report and consider the number of birds sampled by species in a particular study compound. While diversity is important, the number of avian hosts sampled for a given species at any given place and time might be expected to influence the probability of detecting MGE-borne genes in E. coli isolated from a particular avian species. This potential bias should I think be considered within the structure of the GLMM. Perhaps this issue is dealt with in some way that I did not appreciate. More information here might help.

I think the authors might usefully note that there is some evidence to suggest that AMR genes persist even in the absence of selective pressure – this has important implications to conclusions made. In particular, it might be useful to provide some indication of how far these bird species move? Are any of them migratory? How might this influence your conclusions and limitations of the study?

JH Response: We agree that home range could have an important influence on the acquisition of MGE's in the absence of selective pressure. As a result, we have re-run all of the GLMMs including an estimate for the home range of each individual bird (based upon allometric scaling to body weight). Home range is not included in any final models. The process of generating allometrically scaled home range estimates is described in the methodology.

Minor points

1. Line 231- This sentence is difficult to understand and should be revised.

JH Response: We have revised this sentence to make it clearer.

2. Lin 126 - Do the genes exert selective pressure?

See response below

3. Further to #2, while a gene might be detected, it may not be phenotypically expressed – this is something that should be noted.

JH Response: Thank you for highlighting the issues raised in points 2 and 3. We have amended this statement so that it now reads “Hence, aside from investigating processes underlying determinism in bacterial genetic diversity, studying the diversity of two sets of genes which might confer adaptive traits to bacteria will enable us to assess whether an association exists between urban land use and the genetic determinants of bacterial selection, with potential implications for human and animal health”.

This is a really splendid piece of work.

Kathleen Alexander

Reviewer #4 (Remarks to the Author):

General statement

Overall this is an interesting paper exploring the impacts of changing land-use as a driver for the emergence of zoonotic disease by modelling the acquisition of virulence and antimicrobial resistance genes in *E. coli* isolated from birds. This is a unique dataset that I believe will be of interest to many. However, I cannot recommend that it is accepted in its current form. Further analyses, particularly of the *E. coli* genomes, and some additional clarification of details would strengthen the stated key findings of this manuscript.

Major comments

1) A key part of the analysis is that the population structure within the *E. coli* isolates was not explored at all. This is important as *E. coli* is a diverse species with different lineages that vary in accessory genome content – a key element of this study. At present, it appears that the *E. coli* isolates are being treated as one largely homogenous group.

JH response: We appreciate the reviewer’s concerns with regards to presenting genetic data, and have included reporting and analysis of MLST in the manuscript.

- While the MLST was determined for the isolates, this was not reported anywhere in the manuscript. While MLST provides only a crude overview of the population structure, these data should be reported.

JH response: We have reported MLST for each *E. coli* isolate in the manuscript (Supplementary Figure 2).

- Given the WGS of the *E. coli* isolates are available, the population structure should be inferred using phylogenetic methods commonly used in bacterial genomics.

JH response: We have used MLST to infer the genetic population structure of *E. coli* in the manuscript. We do not consider that presenting a phylogeny would address the set of hypotheses tested in this study, and have therefore not included it. Sequences from the isolates studied here are intended to contribute to a larger study of *E. coli* genomics (including a full phylogenetic analysis) that will be submitted for publication separately.

- It would be of interest to know if the bacterial population structure correlated with host species at all suggesting possible adaptation to different wild birds. The bacterial population structure may impact the statistical modelling and may need to be addressed. If it does not impact the modelling this should also be addressed.

JH Response: We have performed Fishers Exact test between MLST and host (for the most common STs, present in >5% of isolates), and did not find an association with host type. We also performed Kruskal-Wallis tests between the three most common STs and diversity of accessory genome virulence and AMR genes, and did not find any associations.

On the advice of experienced bacterial population geneticists, we have included a derivative of MLST as a random effect in the models to account for population structure of *E. coli*. Due to the high diversity of STs present in this dataset (and thus large number of singletons), we were unable to account for the relationship between MGE diversity and bacterial population structure by including MLST on the basis of 7 shared loci as a random effect in the models. Instead, we performed a “supercluster” analysis of MLST genes (based on 3 shared loci) which resulted in 5 groups and just 17 singletons. The models were re-run with the different levels of “superMLST” assigned as a random effect, and we have revised the manuscript accordingly. The results for virulence genes have changed (at the avian functional-group level), but the results for AMR genes remain unchanged. We do not feel that these changes have a large impact on the overall results being presented in the manuscript.

We validated this “superMLST” approach in two ways, using high resolution SNP data:

1. Isolates which had identical sequences (at the SNP level), and the same MGE profiles, were collapsed into a single group, and the “superMLST” models described above were re-run. The results were unchanged.
2. A Principal Component Analysis (PCA) was performed on a matrix of the SNP distances between pairs of isolates (an approach that has been used to account for bacterial population structure in GWAS models). The first three principal components were included as fixed effects in each model (as chosen using a broken stick model), giving similar results to the “superMLST” approach described above.

Validatory steps 1 and 2 are not included in the revised manuscript, but demonstrate that the “superMLST” approach to accounting for bacterial populations structure in these models is robust.

- 2) Co-occurrence of mobile genetic elements. The co-occurrence of both virulence genes and AMR genes was not explored in this study. This is important as the movement of genes

together on mobile genetic elements means not all genes detected may be treated as independent acquisition events which may impact the modelling.

- For example, several of the virulence genes detected are all located on the locus of enterocyte effacement (LEE) pathogenicity island (*tir*, *eae*, *espA*, *espB*, *espF* etc). Was there any effort to account for genes that would likely move together on the same pathogenicity island or plasmid (in the case of *bfpA* and *perA*)?
- The co-occurrence of AMR genes should also be considered. E.g. *strA/strB* mediate resistance to streptomycin and are usually co-mobilised together. Does this impact the statistical modelling? I suspect that it would.
- It may be of interest to explore the presence of different plasmid replicons with common AMR genes. The plasmid replicons were screened for already but the data not reported anywhere.

JH Response: Having consulted expert statisticians who study microbial ecology at the University of Edinburgh and University of Oxford, we are not aware of a method that would directly account for correlation between individuals (i.e. genes) within samples in the response variable of our models. Non-independence of the community data that underlies richness counts is a common feature of community ecology where species (or in this case genes) respond to environmental filters, and the GLMMs that we use assume independence of each observation (i.e. avian *E. coli*), but not of individuals within each observation (i.e. the genes present within each *E. coli*). As such, patterns of gene co-occurrence should not violate any statistical assumptions of our models.

However, we agree with the reviewer's concerns that genes co-mobilised on the same MGE might not represent independent acquisition events. To address this as best we can without having access to long-read sequencing (which would enable identification of the location of genes on plasmids), we have re-run the models combining all pairs of genes with 100% co-occurrence (e.g. *bfpA* and *perA*). We present these updated models in the paper, but find that our results remain unchanged. Details of this are added to the methods section.

3) More details of the genomic analyses are required for the virulence and AMR screens which utilised spades-generated assemblies. What was the minimum coverage and minimum identify used? It is important these thresholds are defined. For example, genes in the VF database that are encoded on the LEE were not detected in all isolates. E.g. while *tir* and *espA* were detected in 12 isolates, *eae* was not detected in 2/12 isolates and *espB* was not detected in any of the isolates. This suggests that some of the genes may be missed in the screen possibly due to thresholds applied or perhaps that several of the genes are known to have sequence diversity (there are several papers exploring the diversity of the LEE) and the different variants of these genes may not be in the CGE database. Please comment on these issues.

JH response: As requested we have included further details on the thresholds for AMR and Virulence gene detection.

Line 502 now reads: "*The threshold of AMR gene detection was set to 90% identity and 60% coverage, as this is shown to be the optimal threshold for this method. A 60% coverage threshold was used to ensure that AMR genes spread over two contigs, and/or located on the edge of the contig, were not missed⁴⁵. Virulence genes were identified using VirulenceFinder with 90% minimum match and 60% minimum length.*"

4) While not the focus of this study, it would be of interest to summarise the AMR patterns detected in these *E. coli* isolates for example, by drug class over the sampling period from 2012-2017. Was there an increase in AMR in general or in any specific class over this sampling period? Are there any data on the use of antimicrobials in Nairobi, Kenya? This would be of broad interest as there is little AMR data from commensal *E. coli* isolates.

JH response: Whilst we agree with the reviewer that it would be of interest to look at longitudinal AMR patterns across Nairobi, the *E. coli* isolates were collected over a period of 1 year, rather than 5 years. This detail has been added to the methods section, to avoid any confusion. Data on the use of antimicrobials in Nairobi, Kenya, and epidemiology of AMR phenotypes, has been collected as part of the UrbanZoo study and are being presented in other manuscripts.

5) Fig 1. It was difficult to see the 95% confidence intervals – perhaps split out into four panels to make easier to interpret. The diversity of virulence genes on the y axis – is this based on counts of total number of virulence genes? The same applies for the species richness of avian community in households on the x axis – is this counts of different wild birds detected at a site? Please give n for the number of *E. coli* genomes in the analysis and make the x and y axis labels more informative.

JH response: Thank you for these suggestions. We have split Figure 1 into separate panels, as suggested, added the number of *E. coli* genomes in the analysis, and made the labelling of both X and Y axes clearer.

6) Fig 2. Please give n for the number of *E. coli* genomes in the analysis. Please give further information to interpret the x axis for human density. It is unclear what e.g. 0.5 human density meant.

JH response: Thank you for these suggestions. We have made the labelling of both X and Y axes more informative.

7) It was unclear what was meant by lines 285-290 and why there was a shift to viruses when the focus of the paper is *E. coli*. The emergence of different pathogenic lineages of *E. coli* have been shown many times through the acquisition of virulence determinants in the accessory genome (e.g. on pathogenicity islands or on plasmids).

JH Response: We appreciate the reviewer's comment and agree that the shift to viruses might be confusing to the reader. As suggested, we have annotated the revised manuscript to include a sentence on the acquisition of virulence determinants leading to the emergence of pathogenic *E. coli*.

8) Please provide further details on the QC and filtering of genomes in Methods (Next-generation sequencing paragraph). Please expand upon standard quality control checks of the short read data or provide a reference. E.g. did this include using the phred score?

JH Response: We have now included additional information regarding quality control and filtering and the text reads as follows:

“150 base-pair paired-end reads were generated and short-read WGS data were pre-processed using an automated protocol developed by the Modernising Medical Microbiology

Oxford (MMM) Group to: (i) perform standard quality control checks using fastQC with default settings; (ii) trim reads to remove remnant adaptor sequences using bbduk (parameters: minoverlap=12, k=19, mink=12, hdist=1, ktrim=r) and (iii) perform a Kraken speciation analysis against an in-house database downloaded from the NCBI sequence read archive (www.ncbi.nlm.nih.gov/sra/), with an automated step for removal of contaminant (non-bacterial) reads.”

We now also include details of the parameters used in SPAdes, including the phred offset used, which was 33.

9) It was unclear as what was the final number of *E. coli* genomes included in this study and methods for selecting the final isolates lacked detail. While 242 is given as the number of isolates with either AMR or VF genes detected, it was not clear if this was all the genomes included or a subset of those included.

- Please provide further details and numbers of *E. coli* genomes in the Next Gen Sequencing section that resulted in the final number. As it currently reads I understand that there were 547 strains subject to WGS.

JH Response: A total of 547 faecal samples were collected, and E. coli was presumptively isolated from 274 of these. We have clarified this in line 137, which now reads “274 E. coli isolates, each of which originated from a different individual avian host, were sequenced. Once sequenced, twenty three isolates were removed for being non-E. coli, and ten were removed for having a genome size larger than 6 megabases. As such, a total of 241 E. coli WGS were considered for further analysis.”

- Please provide further detail as to how human and virus read removal was undertaken (line 459).

JH Response: This sentence refers to a step included in the automated Oxford pipeline by which potentially contaminant, non-bacterial reads (e.g reads classified against the database as “human” or “HIV”) reads are removed following the Kraken speciation analysis. We have now amended the wording in the manuscript as follows to make this clearer:

“perform a Kraken speciation analysis with an in-house database downloaded from the NCBI sequence read archive (www.ncbi.nlm.nih.gov/sra/) with an automated step for removal of contaminant (non-bacterial) reads”

- Please provide details as to how isolates were deemed non-*E. coli* (lines 461-462). What was the number excluded and on what basis – e.g on mapping results to AE014075.1?

*JH Response: The mapping of isolates to AE014075.1 was undertaken as part of a wider project, of which the wildlife isolates analysed here comprise a small part. Information about mapping is therefore not relevant to the results presented in this paper and we have removed this to avoid confusion. For this manuscript, speciation analysis was performed using a kmer based approach as part of the Centre for Genetic Epidemiology pipeline, and we have provided additional details regarding this as shown below. Only isolates deemed to be *E. coli* on this basis were taken forward for further analysis.*

*“The assemblies were run through the batch upload mode of the Centre for Genetic Epidemiology web interface hosted by the Technical University of Denmark (<https://cge.cbs.dtu.dk/services/cge/>) which performs speciation analysis⁴², multi-locus sequence typing (MLST)⁴³, detection of resistance genes⁴⁴, detection of virulence genes⁴⁵ and detection and typing of plasmids⁴⁶. Samples deemed as non-*E. coli* on the basis of the speciation analysis with kmerFinder in the Centre for Genetic Epidemiology pipeline were excluded from further analysis.”*

- What was the number of potentially mixed *E. coli* isolates based upon genome assemblies? It would be useful to provide a supplementary table of the QC and assembly metrics of the final *E. coli* isolates included in this study.

JH Response: As mentioned above, we have now included the following text in line 139 of the results – “...ten potentially mixed isolates were removed for having a genome size larger than 6 megabases.”

We have also included a supplementary datasheet (see supplementary materials) with genome size, species, contigs and n50 for all 274 sequenced isolates.

- Were there isolates that passed all the filtering steps but had no AMR or VF genes detected and were not included in the study? If so, what was the number?

JH Response: All isolates for which AMR or VF genes were not detected were included in the analysis. We consider the absence of either AMR or VF genes to be an important component of diversity. To make this clearer we have added the sentence “Isolates for which AMR or virulence genes were not detected were included in these analyses” into the statistical analyses section of the methods. We have also specified the number of isolates which did not carry AMR or VF genes (see line 142).

- In the AMR and VF tables provided – there were 241 not 242 isolates. Is the number 241?

JH Response: Thank you for highlighting this. The total number of sequenced isolates included in the study was 241, as depicted in the AMR and VF tables. We have corrected the total number of isolates in the text, so that it correctly reads 241 instead of 242.

11) The short read data and genome assembly data for the *E. coli* should be made publicly available on a BioProject. This would facilitate others to be able to use and reproduce these data in the future.

*JH Response: The *E. coli* short read data and genome assembly data will be made publicly available on an NCBI BioProject within the next 12 months, once genomic analysis currently being conducted as part of the wider UrbanZoo project has been completed.*

12) While the AMR and VF data are available at an open access repository held by the University of Liverpool, the specific data available should be stated in the manuscript as it was not initially clear that the AMR and VF data was available.

JH Response: We have amended the manuscript, so that it reads “*Data (AMR and virulence gene datasets, and accompanying metadata) are available via an open access repository held by the University of Liverpool*”.

Minor comments

1) Please add the n value for the number of E. coli isolates in this study to the abstract.

JH Response: This has been added to the abstract.

2) Please check italicisation of E. coli (line 120) and that E. coli is spelt out in full in the introduction at first use (line 97 and not line 107).

JH Response: These changes have been made.

2) Please use acronyms after first use (e.g. line 135 whole genome sequences).

JH Response: The revised manuscript has been edited to include acronyms (e.g. WGS and AMR) throughout.

Reviewer #1 (Remarks to the Author):

To the best of my knowledge, nearly all reviewer comments have been adequately addressed. This article will be a useful contribution to the land use and disease ecology literature and should generate discussion. The additional error I can find is that on figure 1, the panel headings and labels/colors for taxon/functional group appear to be mismatched. However, i defer to reviewer 4 and editors regarding the comments on the additional genetic analyses as this is outside of my area of expertise.

Reviewer #4 (Remarks to the Author):

I have reviewed the revised version of this paper and the responses to the initial critique and comments. I appreciate the additional information and analyses undertaken by the authors and would like to thank the authors for their responses which have addressed my criticisms.

One minor comment regarding Fig 1- it looks like the Figure legend and the titles in grey for each plot may not match. E.g Invertebrate and Fruit/Nectar in the legend have the colours for the other.

REVIEWERS' COMMENTS:

Reviewer #1 (Remarks to the Author):

To the best of my knowledge, nearly all reviewer comments have been adequately addressed. This article will be a useful contribution to the land use and disease ecology literature and should generate discussion. The additional error I can find is that on figure 1, the panel headings and labels/colors for taxon/functional group appear to be mismatched. However, i defer to reviewer 4 and editors regarding the comments on the additional genetic analyses as this is outside of my area of expertise.

JH response: Thank you for spotting this error – it has been rectified.

Reviewer #4 (Remarks to the Author):

I have reviewed the revised version of this paper and the responses to the initial critique and comments. I appreciate the additional information and analyses undertaken by the authors and would like to thank the authors for their responses which have addressed my criticisms.

One minor comment regarding Fig 1- it looks like the Figure legend and the titles in grey for each plot may not match. E.g Invertebrate and Fruit/Nectar in the legend have the colours for the other.

JH response: Thank you for spotting this error – it has been rectified.